# Frequency of Follow-Up Attendance and Blood Glucose Monitoring in Type 2 Diabetic Patients at Moderate to High Cardiovascular Risk: A Cross-Sectional Study in Primary Care

**DOI:** 10.3390/ijerph192114175

**Published:** 2022-10-30

**Authors:** Yunyi Li, Qiya Zhong, Sufen Zhu, Hui Cheng, Wenyong Huang, Harry H. X. Wang, Yu-Ting Li

**Affiliations:** 1School of Public Health, Sun Yat-Sen University, Guangzhou 510080, China; 2State Key Laboratory of Ophthalmology, Zhongshan Ophthalmic Center, Sun Yat-Sen University, Guangzhou 510060, China; 3Nuffield Department of Primary Care Health Sciences, University of Oxford, Oxford OX2 6GG, UK; 4Guangdong Provincial Key Laboratory of Ophthalmology and Visual Science, Zhongshan Ophthalmic Center, Sun Yat-Sen University, Guangzhou 510060, China

**Keywords:** follow-up attendance, blood glucose monitoring, type 2 diabetes mellitus, diabetes management, primary care

## Abstract

Regular follow-up attendance in primary care and routine blood glucose monitoring are essential in diabetes management, particularly for patients at higher cardiovascular (CV) risk. We sought to examine the regularity of follow-up attendance and blood glucose monitoring in a primary care sample of type 2 diabetic patients at moderate-to-high CV risk, and to explore factors associated with poor engagement. Cross-sectional data were collected from 2130 patients enrolled in a diabetic retinopathy screening programme in Guangdong province, China. Approximately one-third of patients (35.9%) attended clinical follow-up <4 times in the past year. Over half of patients (56.9%) failed to have blood glucose monitored at least once per month. Multivariable logistic regression analysis showed that rural residents (adjusted odds ratio [aOR] = 0.420, 95% confidence interval [CI] = 0.338–0.522, *p* < 0.001, for follow-up attendance; aOR = 0.580, 95%CI: 0.472–0.712, *p* < 0.001, for blood glucose monitoring) and subjects with poor awareness of adverse consequences of diabetes complications (aOR = 0.648, 95%CI = 0.527–0.796, *p* < 0.001, for follow-up attendance; aOR = 0.770, 95%CI = 0.633–0.937, *p* = 0.009, for blood glucose monitoring) were both less likely to achieve active engagement. Our results revealed an urban–rural divide in patients’ engagement in follow-up attendance and blood glucose monitoring, which suggested the need for different educational approaches tailored to the local context to enhance diabetes care.

## 1. Introduction

Prevention and control of diabetes and its complications represents a formidable public health challenge worldwide. Current estimates suggest that nearly 130 million people are living with diabetes in China, which accounts for 12.8% of the Chinese adult population [1]. As an independent risk factor for cardiovascular disease (CVD), type 2 diabetes mellitus (T2DM) often coexists with other CV risk factors, such as hypertension, obesity, and dyslipidaemia [2]. Diabetes not only harms patients’ physical well-being but also causes serious complications, which has imposed a considerable economic burden in China [3].

The pathogenesis and progressive nature of diabetes requires patients’ inputs to continuous care, regular monitoring of blood glucose, and adherence to tailored medications to control hyperglycaemia. It is therefore particularly important to ensure the routine delivery of effective and affordable healthcare services to diabetic patients. According to the Chinese national standards for delivering basic public service (third edition), people diagnosed with T2DM are provided with regular follow-up care and fasting plasma glucose monitoring on a free-of-charge basis at primary care facilities [4]. Patients are also encouraged to actively engage in diabetes education and annual check-ups.

Regular follow-up care and routine blood glucose monitoring are essential in diabetes management. The results of blood glucose monitoring are important information for physicians to provide appropriate treatment regimens with tailored lifestyle advice to achieve glycaemic control and prevent disease progression [5,6,7]. A large body of evidence has accumulated over the years documenting that regular follow-up care [8,9] and structured monitoring of blood glucose [10,11] could significantly prevent the deterioration in glycaemic status. However, previous investigations showed that more than two-fifth of patients with T2DM failed to achieve follow-up attendance at least once per quarter at community health centres [12], and that only less than one-fifth of T2DM patients were able to perform blood glucose monitoring on a regular basis [13].

Existing studies reported barriers to follow-up attendance at primary care facilities among Chinese T2DM patients. A cross-sectional study conducted in eastern China found that determinants of infrequent follow-up visits included lower household income, absence of health insurance, and lack of telephone communication and community outreach services [12]. A multi-site survey conducted earlier by our team demonstrated that higher education level of physicians, increased volume of daily patients seen, and no provision of home visits acted as risk factors for non-attainment of the target frequency of follow-up care for T2DM [14]. Patient-level factors associated with poor monitoring of blood glucose were also reported elsewhere [13,15,16].

However, there is limited evidence from studies conducted among T2DM patients at moderate to high CV risk, who account for a large proportion of diabetic patients seen in primary care practice and are more likely to experience major adverse outcomes. We therefore sought to examine the regularity of follow-up attendance and blood glucose monitoring in a primary care sample of T2DM patients at moderate to high CV risk, and to explore factors associated with patients’ poor engagement in routine diabetes care.

## 2. Materials and Methods

### 2.1. Study Design

This was a cross-sectional study conducted in an urban–rural fringe in Guangdong province, southern China. The study was part of a larger project on the screening for diabetic retinopathy (DR) in collaboration with the Zhongshan Ophthalmic Center, Sun Yat-Sen University. The screening cohort was originally designed to evaluate the prevalence, incidence, and progression rate of DR in T2DM patients. Diabetic patients enrolled in family doctor teams for routine diabetes management at local primary care facilities were invited for ophthalmoscopic exams. A semi-structured questionnaire was used to collect information on follow-up attendance and blood glucose monitoring among diabetic patients in the 2019 wave of DR screening.

### 2.2. Setting and Data Source

Data were collected at community and township health centres in the primary care settings. All subjects were interviewed face-to-face by trained clinical staff during on-site examination for DR screening. All completed questionnaires were scrutinised by a senior supervisor on a daily basis as a quality control measure. Information were gathered using a semi-structured questionnaire which included: (1) the patient’s basic information, including age, gender, place of residence, education level, marital status, and living relationships; (2) routine lifestyle behaviours, including smoking and drinking status; (2) disease-related information, including duration of diabetes, use of glucose-lowering medications, presence of comorbidity, as well as awareness of adverse consequences of diabetes complications; (4) health services utilisation, including follow-up attendance and blood glucose monitoring; and (5) the patient’s self-rated health scores.

### 2.3. Participants

Study subjects were drawn from one community health centre and nine township health centres in an urban–rural fringe. Patients with T2DM at moderate to high (including very high) CV risk who participated in the 2019 wave of DR screening were included in the present study. Diabetes was diagnosed as having a fasting plasma glucose level ≥ 7.0 mmol/L or glycated haemoglobin (HbA1c) ≥ 6.5% [17]. The presence of diabetes was determined by the attending physician. The target subjects were T2DM patients who had systolic BP ≥ 130 mmHg or diastolic BP ≥ 85 mmHg in accordance with the clinical guideline [18]. The study flowchart was shown in Figure 1.

### 2.4. Study Variables and Measurements

Weight and height were measured by trained medical staff according to a standardised protocol. Body weight was measured with light clothing. Body mass index (BMI) was calculated as weight in kilograms divided by squared height in meters (kg/m^2^). Blood pressure (BP) was measured in a seated position by routinely validated automatic sphygmomanometers. The arm with the higher BP values was obtained following the standard clinical operating procedure.

The dependent variables were the frequency of patients’ follow-up attendance at primary care facilities and blood glucose monitoring. Patients were asked “How many times did you visit the physician at the community or township health centre in 2018?” to collect information about follow-up attendance in the past year. The frequency of follow-up attendance was divided into <4 and ≥4 times per year, according to Chinese national standards (3rd edition) for delivering basic public health services [4]. Answers to the question “How often did you monitor your blood glucose?” were classified into five categories, i.e., never, only when not feeling well, 1–2 times a quarter, 1–2 times a month, and at least 1–2 times a week. We further divided the frequency into once or more per month vs. less than once per month [15], in the binary logistic regression analysis.

Independent variables included demographics, such as age (<65 or ≥65 years), gender (male or female), place of residence (urban or rural), marital status (married or others, with others including single, divorced, and widowed), living relationships (living alone or others), education level (no formal education or primary school and higher); routine lifestyle behaviours including smoking status (current smoking or others, with others including never smoking and past smoking) and regular drinking (regular drinking or others, with others including never drinking and past drinking); and disease-related variables including duration of diabetes (<10 or ≥10 years), use of glucose-lowing medications (yes or no), the presence of comorbidity (yes or no). Healthcare-related variables included venues for blood glucose testing (only at primary care facilities or others; primary care facilities referred to community health centres or township health centres, and other venues referred to home, pharmacy, village clinics, or secondary/tertiary-level hospitals), self-rated health scores (<3 or ≥3 in a range of 1 to 5 where higher scores representing better self-reported health status), overweight or obese (defined as BMI ≥ 24 kg/m^2^ [19]) and poor awareness of adverse consequences of diabetes complications (i.e., a negative answer to the question “do you know any of the adverse consequences of diabetes complications?”).

### 2.5. Statistical Analysis

Double-entry verification was performed in EpiData 3.1 (Odense, Denmark). Descriptive statistics were used to describe subject characteristics, frequency of follow-up attendance and the regularity of blood glucose monitoring. Independent *t*-tests or chi-square tests, where appropriate, were used to compare the characteristics of urban vs. rural patients. Univariate logistic regression analysis was performed to explore the relationship between each independent variable and frequency of follow-up attendance and blood glucose monitoring, respectively. Variables that were significant were further entered into the multivariable binary logistic regression model. Data analysis was conducted using IBM SPSS Statistics 26. A *p*-value less than 0.05 was considered statistically significant.

### 2.6. Ethics Consideration

All participants provided written informed consent. Data anonymisation was carried out by removing all patient identifiers from the dataset before data analysis. Ethics approval was granted by ethics committees at Zhongshan Ophthalmic Center, Sun Yat-Sen University (Ref: 2017KYPJ094) in accordance with the Declaration of Helsinki 2013.

## 3. Results

### 3.1. Characteristics of Study Participants

The study enrolled a total of 2130 eligible patients, of which, 885 were from urban areas and 1245 were from rural areas. Over half of patients were under 65 years of age (51.5%) and were female (55.6%). Less than one-fifth (16.2%) of participants had no formal education. The prevalence of active smoking (17.7%) and regular drinking (12.6%) was low. More than half of participants were overweight or obese. A substantial proportion of patients were diagnosed with diabetes within the past 10 years (78.7%), reported the presence of comorbidity (81.1%), and received glucose-lowering medications (82.3%). More than half (58.8%) of patients had poor awareness of adverse consequences of diabetes complications. We found that 37.9% of patients had their blood glucose tested only at primary care facilities. When compared to urban patients, rural subjects were younger, had less formal education with a higher prevalence of active smoking and regular drinking, had higher self-rated health scores, and had shorter duration of diagnosed disease with a poor awareness of adverse consequences of diabetes complications, but were as likely as urban patients to develop diabetes comorbidity (Table 1).

### 3.2. Frequency of Follow-Up Attendance and Blood Glucose Monitoring

Of 2130 participants in total, 64.1% attended clinical follow-up at the community health centres or township health centres at least four times in the past year, while slightly more than one-third of patients did not achieve the target frequency of follow-up attendance for T2DM. With respect to the frequency of blood glucose monitoring, 13.6% of patients tested their blood glucose at least 1 to 2 times a week, and 51.3% of patients tested 1 to 2 times a quarter. We found a small proportion of patients who tested their blood glucose only when not feeling well and patients who did not have blood glucose tested in the past year. Overall, nearly half (43.1%) of patients were able to have their blood glucose tested once or more per month (Table 2).

### 3.3. Factors Associated with the Frequency of Follow-Up Attendance

In the univariate analysis, age, place of residence, smoking status, self-rated health scores, duration of diabetes, awareness of adverse consequences of diabetes complications, use of glucose-lowering medications, and venues for testing blood glucose were associated with the frequency of follow-up attendance (*p* < 0.05), which were further entered in the multivariable analysis. In the further analysis, patients who received glucose-lowering medications (aOR = 2.319, 95%CI: 1.782–3.019; *p* < 0.001) or had blood glucose tested only at primary care facilities (aOR = 5.161, 95%CI: 4.095–6.503; *p* < 0.001) were more prone to have a higher regularity of follow-up attendance compared to their counterparts. However, patients who lived in rural areas (aOR = 0.420, 95%CI: 0.338–0.522; *p* < 0.001), had poor awareness of adverse consequences of diabetes complications (aOR = 0.648, 95%CI: 0.527–0.796; *p* < 0.001) with self-rated health scores ≥3 (aOR = 0.665, 95%CI: 0.509–0.868; *p* = 0.003) were less likely to attend clinical follow-up at least 4 times in the past year (Figure 2).

### 3.4. Factors Associated with the Frequency of Blood Glucose Monitoring

In the univariate analysis, age, place of residence, education level, self-rated health scores, duration of diabetes, presence of comorbidity, awareness of adverse consequences of diabetes complications, use of glucose-lowering medications, and venues for testing blood glucose were associated with the frequency of blood glucose monitoring (*p* <0.05), which were further entered in the multivariable analysis. In the further analysis, the following factors remained significantly associated with having blood glucose monitored once or more per month: age ≥ 65 years (aOR = 1.241, 95%CI: 1.020–1.509; *p* = 0.031), living in rural areas (aOR = 0.580, 95%CI: 0.472–0.712; *p* < 0.001), having formal education (aOR = 1.529, 95%CI: 1.171–1.996; *p* = 0.002), having a diagnosis of diabetes for over 10 years (aOR = 1.415, 95%CI: 1.120–1.787; *p* = 0.004), poor awareness of adverse consequences of diabetes complications (aOR = 0.770, 95%CI: 0.633–0.937; *p* = 0.009), use of glucose-lowering medications (aOR = 2.538, 95%CI: 1.933–3.334; *p* < 0.001), and having blood glucose tested only at primary care facilities (aOR = 0.241, 95%CI: 0.196–0.297; *p* < 0.001) (Figure 3).

## 4. Discussion

### 4.1. Main Findings

We found that approximately one-third of patients (35.9%) attended clinical follow-up <4 times in the past year, and that slightly over half of patients (56.9%) failed to monitor their blood glucose at least once per month. Multivariable logistic regression analysis showed that place of residence, awareness of consequences of diabetes complications, use of glucose-lowering medications, and venue for testing blood glucose were factors associated with the regularity of both follow-up attendance and blood glucose monitoring. Moreover, current smoking and having better self-rated health status were negatively associated with the attainment of the recommended frequency target for follow-up visits at primary care facilities. Patients aged 65 years or older, having an education level of primary school and above, or having a diagnosed diabetes for over 10 years were more likely to have their blood glucose monitored once and above per month. Our results exhibited an urban–rural divide in patients’ engagement in follow-up attendance and blood glucose monitoring.

### 4.2. Relationship with Other Studies

In China, the management of T2DM has been integrated as part of the national basic public health service delivery [20]. Follow-up care is recommended to be provided to patients with T2DM on a quarterly basis according to the Chinese national standards (third edition) for delivering basic public health service [4]. In the present study, we found that up to two-thirds (64.1%) of T2DM patients at moderate to high CV risk attended clinical follow-up at least four times per year. The proportion was consistent with results from another cross-sectional survey conducted in western China [21], but was lower than that reported in a longitudinal study in Luxembourg, Belgium [22], where 90% of T2DM patients consulted physicians at least 4 times a year. Another cross-sectional study among the Norwegian population found that high adherence to recommended diabetes follow-up procedures in primary care was associated with better glycaemic control and lower estimated CV risk in diabetic patients [23].

Multivariable analysis revealed that rural residence, current smoking, higher self-rated health scores, lack of awareness of adverse consequences of diabetes complications, absence of drug use, and having mixed venues for blood glucose monitoring were factors associated with less regularity of follow-up attendance among study subjects. The results were consistent with our expectations and existing literature which shows that poor primary care gate-keeping is more common in deprived neighbourhoods [24]. A systematic review found that smoking and poor knowledge of diabetes complications were factors associated with follow-up non-attendance [25]. Our observation that patients who were not on glucose-lowering medications had fewer follow-up visits to primary care physicians was also similarly reported in other studies [22,26,27]. It is possible that patients who received antihyperglycaemic drugs were given greater attention through education programmes, personal instructions, and screening. Patients with a better self-rated health status were less likely to regularly attend clinical follow-up in the present study. Results from the Danish arm of the ADDITION-study did not report a significant association between self-rated health and attendance at clinical follow-up [28], but demonstrated that non-adherence to follow-up may occur if patients did not experience a need for extra care because of being relatively healthy. In the context of the Chinese basic public health service delivery, a free-of-charge blood glucose test on-site is available on a quarterly basis for T2DM patients attending primary care facilities [4]. In our study, it is not surprising to observe that patients who had their blood glucose tested only at primary care centres were more likely to achieve greater follow-up attendance.

Routine blood glucose monitoring helps provide timely information on blood glucose level of diabetic patients, in addition to diet, exercise, and medication adherence profiles, to inform evidence-based clinical decision-making. Fasting plasma glucose monitoring is recommended for patients with T2DM on a quarterly basis in primary care diabetes management [4]. In our study, 94.4% of patients with T2DM at moderate to high CVD risk monitored their blood glucose more than once per quarter, which was higher than that reported (88.4%) in another cross-sectional study conducted in Shandong, eastern China [16]. We found that most of patients utilised blood glucose testing services provided by primary care facilities. It is recommended that T2DM patients with lifestyle interventions or on oral hypoglycaemic drugs should have blood glucose monitoring at least twice a week on average [7]. However, the proportion of patients who had weekly blood glucose monitoring in the present study was much lower than that reported in western countries [29,30]. This implies a poor engagement in diabetes care among T2DM patients in China, particularly among those at moderate to high CV risk. 

Earlier findings from a cross-sectional survey in eastern China suggested that diabetic patients who lived in urban areas, had a higher education level, had the presence of physician-diagnosed diabetes for over 10 years, and had insulin injection tended to achieve a higher frequency of blood glucose monitoring [15]. This was similarly observed in our study. We further found that a higher self-rated health status was associated with less frequent monitoring of blood glucose, although such association was attenuated after adjustment. It is possible that the motivation to perform blood glucose monitoring is largely influenced by the perceived concern on disease progression [31], and thus a poor self-rated health status was often associated with higher HbA1c levels [32,33]. Therefore, frequent monitoring of blood glucose among patients with poor self-rated health status may be explained by the fear of adverse consequences of diabetic hyperosmolar syndrome. Our results showed that poor awareness of consequences of diabetes complications was associated with less frequent monitoring of blood glucose. It was previously reported that a lower frequency of blood glucose monitoring profile was related to poor knowledge of blood glucose control [16]. It is reasonable to assume that patients who reported that they had their blood glucose tested only at primary care facilities may have no additional monitoring of blood glucose or blood pressure at home, which may contribute to the barriers to continuous treatment [34]. This may underscore a greater need for developing patients’ knowledge and skills required to control blood glucose among T2DM patients with risk factors identified in our study.

### 4.3. Implications for Research and Practice

The high rate of uncontrolled blood glucose reported in a large-scale survey in China [35] has posed a significant challenge for effective diabetes self-management and appropriate intervention with follow-up strategies, which are part of primary care for addressing multimorbidity [36,37]. Previous studies found that increased primary care visits and blood glucose monitoring could lead to improved glycaemic control and lower the risk of adverse CV outcomes [10,23,38]. Patients with T2DM at higher CV risk often need more intensive care, yet they were found to have poor engagement in follow-up attendance and blood glucose monitoring. In addition, inadequate awareness of adverse consequences of diabetes complications remained common, and was associated with a lower frequency of both follow-up attendance and blood glucose monitoring. Therefore, dedicated and continuous educational inputs from primary care practitioners [39], in combination with effective risk communications [40], are needed in daily clinical practice.

Continuous education on diabetes self-management to facilitate the knowledge and skills required for diabetes care can increase diabetes awareness and improve glycaemic control [41,42], and can exert positive effects on clinical visits and monitoring of blood glucose [43,44,45]. This may be particularly crucial for patients with less access to adequate, structured health education delivered by qualified professionals in primary care settings, compared to that in hospital-based specialist care settings [44]. Education of knowledge on diabetes and its complications should therefore be emphasised during clinical encounter in primary care, accompanied by standardised diabetes education and training programmes especially in rural areas where diabetic patients are less likely to be aware of diabetes complications. Furthermore, the urban–rural divide should be taken into account in diabetes education practice. Rural patients tend to be younger, have less formal education, smoke and drink more, and have shorter diagnosed disease duration, but are just as likely as urban patients to develop diabetes complications as shown in our study. Given the existing evidence on the association between socio-demographic factors and participation in diabetes education [46,47], various educational approaches tailored to the local socio-economic context will be essential to strengthen the capacity of diabetes care. Diabetes education with printed materials may help maximise the effective communication of health messages [48,49], which may be relevant to T2DM patients at higher CV risk who tend to have lower educational attainment in the urban–rural fringe.

### 4.4. Strengths and Weaknesses of the Study

The target subjects in this study consisted of T2DM patients at moderate to high CV risk, who are commonly encountered in daily primary care but are often struggling with barriers to engage in regular follow-up attendance and routine blood glucose monitoring. A semi-structured questionnaire was used to collect a broad range of information, including the patient’s demographics, routine lifestyle behaviours, disease-related information, healthcare utilisation, and self-reported health status through face-to-face interviews by trained medical practitioners. However, this study has some limitations that should be considered. First, we did not have an electronic health record system where data on follow-up attendance and blood glucose monitoring can be directly retrieved. Thus, recall bias may arise from self-reporting despite rigorous quality control. Secondly, the awareness of adverse consequences of diabetes complications was ascertained based on a single question. This may limit the extent to which our results are comparable with other studies due to inconsistent methodology. Third, subjects in this study were drawn from the register of diabetic management in primary care settings. Patients with T2DM who were not enrolled may not be captured in our study. Last but not least, we were not able to make inferences about causality or temporal ordering of events due to the cross-sectional nature of the study. Future research using data collected from computerised health records will help to provide longitudinal evidence for optimising diabetes care.

## 5. Conclusions

In the present study, our findings suggested that follow-up care delivered in primary care was not adequately utilised by Chinese patients with T2DM who were at moderate to high CV risk, and that patients’ engagement in blood glucose monitoring was suboptimal overall. Living in rural areas and having poor awareness of diabetes complications acted as significant barriers to both follow-up attendance and blood glucose monitoring. This would require dedicated and continuous educational inputs from primary care practitioners to increase awareness of adverse consequences of diabetes complications in daily clinical practice. Our results revealed a significant urban–rural divide in patients’ engagement in regular follow-up attendance and blood glucose monitoring, which suggested the need for different educational approaches tailored to the local socio-economic context to strengthen the capacity of diabetes care.

## Figures and Tables

**Figure 1 ijerph-19-14175-f001:**
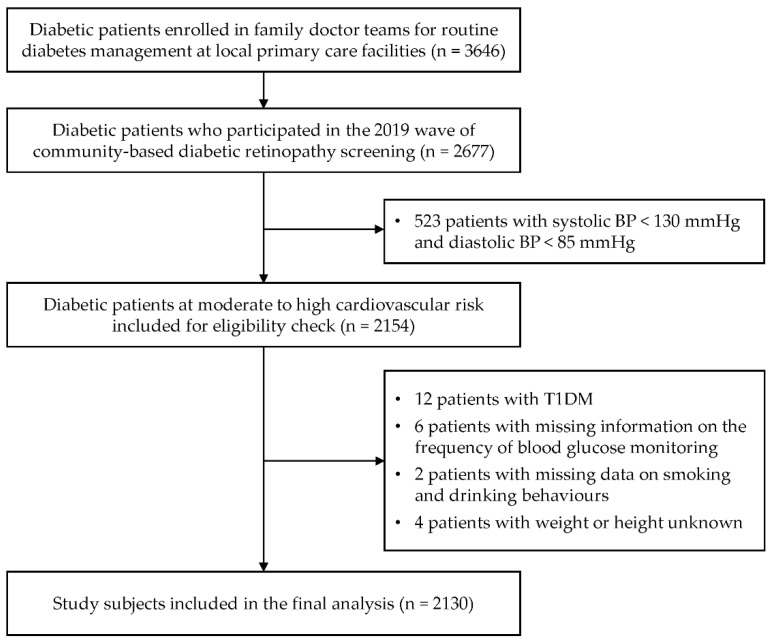
Flow chart of the study. T1DM, type 1 diabetes mellitus; T2DM, type 2 diabetes mellitus; BP, blood pressure.

**Figure 2 ijerph-19-14175-f002:**
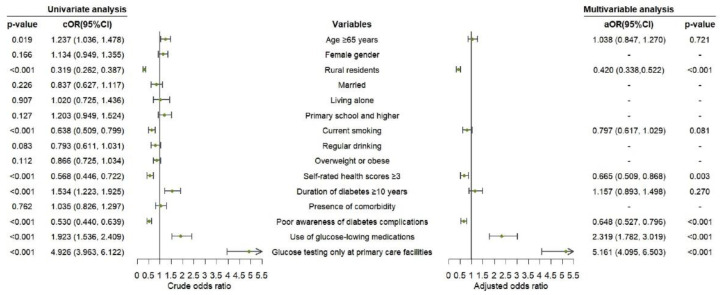
Logistic regression analysis on factors associated with regular follow-up attendance among T2DM patients at moderate to high CV risk.

**Figure 3 ijerph-19-14175-f003:**
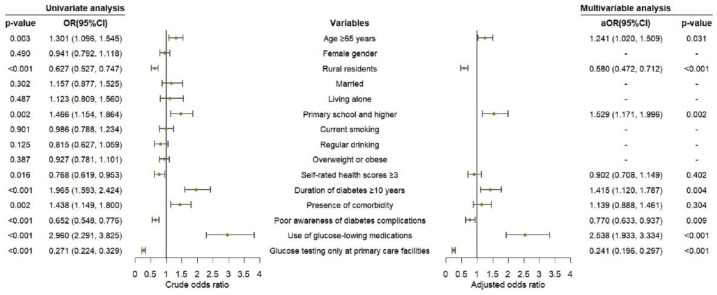
Logistic regression analysis on factors associated with regular blood glucose monitoring among T2DM patients at moderate to high CV risk.

**Table 1 ijerph-19-14175-t001:** Characteristics of study participants.

Variables	Total (n = 2130)	Urban (n = 885)	Rural (n = 1245)	*p*-Value
Age, years				<0.001
<65	1097 (51.5%)	363 (41.0%)	734 (59.0%)	
≥65	1033 (48.5%)	522 (59.0%)	511 (41.0%)	
Gender				0.092
Male	946 (44.4%)	374 (42.3%)	572 (45.9%)	
Female	1184 (55.6%)	511 (57.7%)	673 (54.1%)	
Marital status				0.541
Others	235 (11.0%)	102 (11.5%)	133 (10.7%)	
Married	1895 (89.0%)	783 (88.5%)	1112 (89.3%)	
Living relationships				0.946
Others	1975 (92.7%)	821 (92.8%)	1154 (92.7%)	
Living alone	155 (7.3%)	64 (7.2%)	91 (7.3%)	
Education level				<0.001
No formal education	344 (16.2%)	93 (10.5%)	251 (20.2%)	
Primary school and above	1786 (83.8%)	792 (89.5%)	994 (79.8%)	
Smoking status				<0.001
Others	1752 (82.3%)	784 (88.6%)	968 (77.8%)	
Current smoking	378 (17.7%)	101 (11.4%)	277 (22.2%)	
Drinking status				<0.001
Others	1862 (87.4%)	814 (92.0%)	1048 (84.2%)	
Regular drinking	268 (12.6%)	71 (8.0%)	197 (15.8%)	
Overweight or obese				0.073
No	990 (46.5 %)	391 (44.2%)	599 (48.1%)	
Yes	1140 (53.5%)	494 (55.8%)	646 (51.9%)	
Self-rated health scores				<0.001
<3	411 (19.3%)	214 (24.2%)	197 (15.8%)	
≥3	1719 (80.7%)	671 (75.8%)	1048 (84.2%)	
Duration of diabetes, years				<0.001
<10	1677 (78.7%)	607 (68.6%)	1070 (85.9%)	
≥10	453 (21.3%)	278 (31.4%)	175 (14.1%)	
Presence of comorbidity				0.179
No	402 (18.9%)	179 (20.2%)	223 (17.9%)	
Yes	1728 (81.1%)	706 (79.8%)	1022 (82.1%)	
Poor awareness of consequences of diabetes complications				<0.001
No	878 (41.2%)	447 (50.5%)	431 (34.6%)	
Yes	1252 (58.8%)	438 (49.5%)	814 (65.4%)	
Use of glucose-lowering medications				0.267
No	377 (17.7%)	147 (16.6%)	230 (18.5%)	
Yes	1753 (82.3%)	738 (83.4%)	1015 (81.5%)	
Venue for blood glucose testing				<0.001
Mixed venues	1323 (62.1%)	472 (53.3%)	851 (68.4%)	
Primary care only	807 (37.9%)	413 (46.7%)	394 (31.6%)	

**Table 2 ijerph-19-14175-t002:** Frequency of follow-up attendance and blood glucose monitoring among T2DM patients at moderate to high CV risk.

Variables	N	% (95%CI)
Frequency of follow-up attendance		
Below 4 times per year	765	35.9% (33.9–37.9%)
4 times and above per year	1365	64.1% (62.1–66.1%)
Frequency of blood glucose monitoring		
1–2 times per week	290	13.6% (12.1–15.1%)
1–2 times per month	629	29.5% (27.6–31.4%)
1–2 times per quarter	1093	51.3% (49.2–53.2%)
Only when not feeling well	59	2.8% (2.1–3.5%)
Never	59	2.8% (2.1–3.5%)

## Data Availability

The data presented in this study are available on reasonable request from the corresponding author.

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
