# Peer review of "Frequency of Follow-Up Attendance and Blood Glucose Monitoring in Type 2 Diabetic Patients at Moderate to High Cardiovascular Risk: A Cross-Sectional Study in Primary Care"

_ijerph, 2022, doi:10.3390/ijerph192114175_

Round 1
Reviewer 1 Report
This is a very informative and well-organized and written paper. I just had a few minor comments that need to be addressed:
Although the abstract is not short it is not as informative as it is supposed to be and cannot stand alone, especially in the results section for the regression and you also missed the number of patients. Also, the conclusion was generalized to other populations that may not resemble this population.
The first paragraph is not relevant to the study objective, so you should remove it and get directly into your main idea for the paper. One exception is the last sentence of the first paragraph which can be reorganized with the rest of the introduction to make the first paragraph more informative and relevant to your objective.
Double-check the accuracy of this sentence "frequency of blood glucose monitoring was negatively associated with the glycemic level in patients with T2DM"; as written it infers that patients with more frequent monitoring have poor glycemic control.
Minor points:
Page 2, line 73 "analyzed"; Page 3, line 127 "respectively."; Page 10, Line 325 "Preview", line 332 "was both risk factors associated ... ", and line 336 "DSEM" these are some typo mistakes.
"Multivariate" should be changed to "multivariable" throughout the manuscript.
Author Response
Dear Reviewer 1,
We would like to thank you for your most favourable comments. We have revised our paper according to your valuable comments, with revisions shown in track changes in our manuscript. The point-by-point responses are shown as below.
Point 1: This is a very informative and well-organized and written paper. I just had a few minor comments that need to be addressed:
Response 1: Thank you for your most favourable comments.
Point 2: Although the abstract is not short it is not as informative as it is supposed to be and cannot stand alone, especially in the results section for the regression and you also missed the number of patients. Also, the conclusion was generalized to other populations that may not resemble this population.
Response 2: We appreciate your valuable suggestions. We have re-written the abstract following your expert advice while within the word count limit (200 words) of the journal. The revised Abstract section now reads as follows.
Regular follow-up attendance in primary care and routine blood glucose monitoring are essential in diabetes management, particularly for patients at higher cardiovascular (CV) risk. We sought to examine the regularity of follow-up attendance and blood glucose monitoring in a primary care sample of type 2 diabetic patients at moderate-to-high CV risk, and to explore factors associated with poor engagement. Cross-sectional data were collected from 2130 patients enrolled in a diabetic retinopathy screening programme in Guangdong province, China. Approximately one-third of patients (35.9%) attended clinical follow-up <4 times in the past year. Over half of patients (56.9%) failed to have blood glucose monitored at least once per month. Multivariable logistic regression analysis showed that rural residents (adjusted odds ratio [aOR]=0.420, 95% confidence interval [CI]=0.338-0.522, p<0.001, for follow-up attendance; aOR=0.580, 95%CI: 0.472-0.712, p<0.001, for blood glucose monitoring) and subjects with poor awareness of adverse consequences of diabetes complications (aOR=0.648, 95%CI=0.527-0.796, p<0.001, for follow-up attendance; aOR=0.770, 95%CI=0.633-0.937, p=0.009, for blood glucose monitoring) were both less likely to achieve active engagement. Our results revealed an urban-rural divide in patients’ engagement in follow-up attendance and blood glucose monitoring, which suggested the need for different educational approaches tailored to the local context to enhance diabetes care. (200 /200 words)
Please see the Abstract section for the revision. Thank you.
Point 3: The first paragraph is not relevant to the study objective, so you should remove it and get directly into your main idea for the paper. One exception is the last sentence of the first paragraph which can be reorganized with the rest of the introduction to make the first paragraph more informative and relevant to your objective.
Response 3: We appreciate your valuable suggestions. We have removed the original paragraph as indicated, and also re-organised the entire Introduction section while simultaneously taking into account the comments from the other reviewers. The revised Introduction section now reads as follows:
Prevention and control of diabetes and its complications represents a formidable public health challenge worldwide. Current estimates suggest that nearly 130 million people are living with diabetes in China, which account for 12.8% of the Chinese adult population [1]. As an independent risk factor for cardiovascular disease (CVD), type 2 diabetes mellitus (T2DM) often coexists with other CV risk factors, such as hypertension, obesity, and dyslipidaemia [2]. Diabetes not only harms patients’ physical well-being but also causes serious complications, which has imposed huge economic burden in China [3].
The pathogenesis and progressive nature of diabetes requires patients’ inputs to continuous care, regular monitoring of blood glucose, and adherence to tailored medications to control hyperglycaemia. It is therefore particularly important to ensure the routine delivery of effective and affordable healthcare services to diabetic patients. According to the Chinese national standards for delivering basic public service (third edition), people diagnosed with T2DM are provided with regular follow-up care and fasting plasma glucose monitoring on a free-of-charge basis at primary care facilities [4]. Patients are also encouraged to actively engage in diabetes education and annual check-ups.
Regular follow-up care and routine blood glucose monitoring are essential in diabetes management. The results of blood glucose monitoring are important information for physicians to give appropriate treatment regimen with tailored lifestyle advice to achieve glycaemic control and prevent disease progression [5-7]. A large body of evidence has accumulated over the years documenting that regular follow-up care [8, 9] and structured monitoring of blood glucose [10,11] could significantly prevent the deterioration in glycaemic status. However, previous investigations showed that more than two-fifth of patients with T2DM failed to achieve follow-up attendance at least once per quarter at community health centres [12], and that only less than one-fifth of T2DM patients were able to perform blood glucose monitoring on a regular basis [13].
Existing studies reported barriers to follow-up attendance at primary care facilities among Chinese T2DM patients. A cross-sectional study conducted in eastern China found that determinants of infrequent follow-up visits included lower household income, absence of health insurance, and lack of telephone communication and community outreach services [12]. A multi‑site survey conducted earlier by our team demonstrated that higher education level of physicians, increased volume of daily patients seen, and no provision of home visits acted as risk factors for non‑attainment of the target frequency of follow‑up care for T2DM [14]. Patient-level factors associated with poor monitoring of blood glucose were also reported elsewhere [13, 15-16].
However, there is limited evidence from studies conducted among T2DM patients at moderate to high CV risk, who account for a large proportion of diabetic patients seen in primary care practice and are more likely to experience major adverse outcomes. We therefore sought to examine the regularity of follow-up attendance and blood glucose monitoring in a primary care sample of T2DM patients at moderate to high CV risk, and to explore factors associated with patients’ poor engagement in routine diabetes care.
Please see the Introduction section for the revision. Thank you.
Point 4: Double-check the accuracy of this sentence "frequency of blood glucose monitoring was negatively associated with the glycemic level in patients with T2DM"; as written it infers that patients with more frequent monitoring have poor glycemic control.
Response 4: We appreciate your valuable suggestions. We have re-written the sentence as follows:
A large body of evidence has accumulated over the years documenting that regular follow-up care [8, 9] and structured monitoring of blood glucose [10,11] could significantly prevent the deterioration in glycaemic status.
Please see lines 56-59 for the revision. Thank you.
Point 5:
Minor points: Page 2, line 73 "analyzed"; Page 3, line 127 "respectively."; Page 10, Line 325 "Preview", line 332 "was both risk factors associated ... ", and line 336 "DSEM" these are some typo mistakes.
"Multivariate" should be changed to "multivariable" throughout the manuscript.
Response 5: We thank you for your valuable suggestions, and have made the corrections accordingly throughout the manuscript. Thank you again for all your valuable comments.
Reviewer 2 Report
1. The manuscript should be carefully checked and corrected for grammatical, typographical, and punctuation errors.
2. The manuscript title should be revised.
3. The authors should be revised the abstract and conclusion to emphasize the most crucial findings.
4. Introduction part; The primary task of the introduction is to outline the research background, history, the problems already solved and the achievements already made, and the problems still unsolved and new problems found. That is to say; the introduction is a brief review or summary of the relevant works of literature in the related fields. The authors should make an objective evaluation or comparison of the previous research, pointing out the problems and unsolved technological gaps, and then propose new ways(s) and new ideas(s) to solve one of these problems, thus displaying the motivation and significance of the manuscript's topic.
Author Response
Dear Reviewer 2,
We would like to thank you for your expert and constructive comments. We have revised our paper according to your kind suggestions, with revisions shown in track changes in our manuscript. The point-by-point responses are shown as below.
Point 1: The manuscript should be carefully checked and corrected for grammatical, typographical, and punctuation errors.
Response 1: Thank you for your kind suggestions. We have carefully read through the manuscript to ensure the absence of grammatical, typographical, and punctuation errors throughout the paper.
Point 2: The manuscript title should be revised.
Response 2: We appreciate your valuable suggestions. We have re-written the manuscript title as – Frequency of follow-up attendance and blood glucose monitoring in type 2 diabetic patients at moderate to high cardiovascular risk: a cross-sectional study in primary care. Thank you.
Point 3: The authors should be revised the abstract and conclusion to emphasize the most crucial findings.
Response 3: We appreciate your valuable suggestions. We have re-written the abstract following your expert advice while within the word count limit (200 words) of the journal. The revised Abstract section now reads as follows.
Regular follow-up attendance in primary care and routine blood glucose monitoring are essential in diabetes management, particularly for patients at higher cardiovascular (CV) risk. We sought to examine the regularity of follow-up attendance and blood glucose monitoring in a primary care sample of type 2 diabetic patients at moderate-to-high CV risk, and to explore factors associated with poor engagement. Cross-sectional data were collected from 2130 patients enrolled in a diabetic retinopathy screening programme in Guangdong province, China. Approximately one-third of patients (35.9%) attended clinical follow-up <4 times in the past year. Over half of patients (56.9%) failed to have blood glucose monitored at least once per month. Multivariable logistic regression analysis showed that rural residents (adjusted odds ratio [aOR]=0.420, 95% confidence interval [CI]=0.338-0.522, p<0.001, for follow-up attendance; aOR=0.580, 95%CI: 0.472-0.712, p<0.001, for blood glucose monitoring) and subjects with poor awareness of adverse consequences of diabetes complications (aOR=0.648, 95%CI=0.527-0.796, p<0.001, for follow-up attendance; aOR=0.770, 95%CI=0.633-0.937, p=0.009, for blood glucose monitoring) were both less likely to achieve active engagement. Our results revealed an urban-rural divide in patients’ engagement in follow-up attendance and blood glucose monitoring, which suggested the need for different educational approaches tailored to the local context to enhance diabetes care. (200 /200 words)
Please see the Abstract section for the revision. Thank you.
Point 4: Introduction part; The primary task of the introduction is to outline the research background, history, the problems already solved and the achievements already made, and the problems still unsolved and new problems found. That is to say; the introduction is a brief review or summary of the relevant works of literature in the related fields. The authors should make an objective evaluation or comparison of the previous research, pointing out the problems and unsolved technological gaps, and then propose new ways(s) and new ideas(s) to solve one of these problems, thus displaying the motivation and significance of the manuscript's topic.
Response 4: We appreciate your valuable suggestions. We have re-organised the entire Introduction section following your expert advice, while simultaneously taking into account comments from the other reviewers. The revised Introduction section now reads as follows:
Prevention and control of diabetes and its complications represents a formidable public health challenge worldwide. Current estimates suggest that nearly 130 million people are living with diabetes in China, which account for 12.8% of the Chinese adult population [1]. As an independent risk factor for cardiovascular disease (CVD), type 2 diabetes mellitus (T2DM) often coexists with other CV risk factors, such as hypertension, obesity, and dyslipidaemia [2]. Diabetes not only harms patients’ physical well-being but also causes serious complications, which has imposed huge economic burden in China [3].
The pathogenesis and progressive nature of diabetes requires patients’ inputs to continuous care, regular monitoring of blood glucose, and adherence to tailored medications to control hyperglycaemia. It is therefore particularly important to ensure the routine delivery of effective and affordable healthcare services to diabetic patients. According to the Chinese national standards for delivering basic public service (third edition), people diagnosed with T2DM are provided with regular follow-up care and fasting plasma glucose monitoring on a free-of-charge basis at primary care facilities [4]. Patients are also encouraged to actively engage in diabetes education and annual check-ups.
Regular follow-up care and routine blood glucose monitoring are essential in diabetes management. The results of blood glucose monitoring are important information for physicians to give appropriate treatment regimen with tailored lifestyle advice to achieve glycaemic control and prevent disease progression [5-7]. A large body of evidence has accumulated over the years documenting that regular follow-up care [8, 9] and structured monitoring of blood glucose [10,11] could significantly prevent the deterioration in glycaemic status. However, previous investigations showed that more than two-fifth of patients with T2DM failed to achieve follow-up attendance at least once per quarter at community health centres [12], and that only less than one-fifth of T2DM patients were able to perform blood glucose monitoring on a regular basis [13].
Existing studies reported barriers to follow-up attendance at primary care facilities among Chinese T2DM patients. A cross-sectional study conducted in eastern China found that determinants of infrequent follow-up visits included lower household income, absence of health insurance, and lack of telephone communication and community outreach services [12]. A multi‑site survey conducted earlier by our team demonstrated that higher education level of physicians, increased volume of daily patients seen, and no provision of home visits acted as risk factors for non‑attainment of the target frequency of follow‑up care for T2DM [14]. Patient-level factors associated with poor monitoring of blood glucose were also reported elsewhere [13, 15-16].
However, there is limited evidence from studies conducted among T2DM patients at moderate to high CV risk, who account for a large proportion of diabetic patients seen in primary care practice and are more likely to experience major adverse outcomes. We therefore sought to examine the regularity of follow-up attendance and blood glucose monitoring in a primary care sample of T2DM patients at moderate to high CV risk, and to explore factors associated with patients’ poor engagement in routine diabetes care.
Please see the Introduction section for the revision. Thank you again for all your valuable comments.
Reviewer 3 Report
Overall a well written paper . I feel that the abstract can be improved to make it more "citable"". The intent of the paper is to show that because there are differences in populations, therefore different approaches must be made for diabetes education For example the study findings in Table 1 showed that
a. There was a rural urban divide ; Rural patients are more likely to be younger ( 60% rural vs 40 % urban ), have less formal education, smoke and drink more , and have shorter diagnosed disease duration ( 85% of rural had <10yr duration vs 70% of urban ) but just as likely to have diabetes complications (80% ) .
b. Rural diabetics were less likely to be aware of diabetes complications ( 65% unaware vs 49.5% of urban )
b. Nevertheless, 80% of rural and 90% of urban diabetics had primary and abover education ( so printed materials might still help ? )
Having hard data that others can quote I feel makes your paper more likely to be read and cited , eg by other workers trying to design their own outreach programmes
Author Response
Dear Reviewer 3,
We would like to thank you for your most favourable comments. We have revised our paper according to your valuable comments, with revisions shown in track changes in our manuscript. The point-by-point responses are shown as below.
Point 1: Overall a well written paper.
Response 1: Thank you for your most favourable comments.
Point 2: I feel that the abstract can be improved to make it more "citable"". The intent of the paper is to show that because there are differences in populations, therefore different approaches must be made for diabetes education
Response 2: We appreciate your valuable suggestions. We have re-written the abstract following your expert advice while within the word count limit (200 words) of the journal. The revised Abstract section now reads as follows.
Regular follow-up attendance in primary care and routine blood glucose monitoring are essential in diabetes management, particularly for patients at higher cardiovascular (CV) risk. We sought to examine the regularity of follow-up attendance and blood glucose monitoring in a primary care sample of type 2 diabetic patients at moderate-to-high CV risk, and to explore factors associated with poor engagement. Cross-sectional data were collected from 2130 patients enrolled in a diabetic retinopathy screening programme in Guangdong province, China. Approximately one-third of patients (35.9%) attended clinical follow-up <4 times in the past year. Over half of patients (56.9%) failed to have blood glucose monitored at least once per month. Multivariable logistic regression analysis showed that rural residents (adjusted odds ratio [aOR]=0.420, 95% confidence interval [CI]=0.338-0.522, p<0.001, for follow-up attendance; aOR=0.580, 95%CI: 0.472-0.712, p<0.001, for blood glucose monitoring) and subjects with poor awareness of adverse consequences of diabetes complications (aOR=0.648, 95%CI=0.527-0.796, p<0.001, for follow-up attendance; aOR=0.770, 95%CI=0.633-0.937, p=0.009, for blood glucose monitoring) were both less likely to achieve active engagement. Our results revealed an urban-rural divide in patients’ engagement in follow-up attendance and blood glucose monitoring, which suggested the need for different educational approaches tailored to the local context to enhance diabetes care. (200 /200 words)
Please see the Abstract section for the revision. Thank you.
Point 3: For example the study findings in Table 1 showed that a. There was a rural urban divide ; Rural patients are more likely to be younger ( 60% rural vs 40 % urban ), have less formal education, smoke and drink more , and have shorter diagnosed disease duration ( 85% of rural had <10yr duration vs 70% of urban ) but just as likely to have diabetes complications (80% ). b. Rural diabetics were less likely to be aware of diabetes complications ( 65% unaware vs 49.5% of urban ) b. Nevertheless, 80% of rural and 90% of urban diabetics had primary and abover education ( so printed materials might still help ? ) Having hard data that others can quote I feel makes your paper more likely to be read and cited , eg by other workers trying to design their own outreach programmes
Response 3: We appreciate your valuable suggestions. We have followed your expert advice to revise our abstract, and have also added to the discussion section regarding the implications for research and practice. The 2nd paragraph under the subsection 4.3 Implications for research and practice now reads as follows:
Continuous education on diabetes self-management to facilitate the knowledge and skills required for diabetes care can increase diabetes awareness and improve glycaemic control [41,42], and can exert positive effects on clinical visits and monitoring of blood glucose [43-45]. This may be particularly crucial for patients with less access to adequate, structured health education delivered by qualified professionals in primary care settings, compared to that in hospital-based specialist care settings [44]. Education of knowledge on diabetes and its complications should therefore be emphasised during clinical encounter in primary care, accompanied by standardised diabetes education and training programmes especially in rural areas where diabetic patients are less likely to be aware of diabetes complications. Furthermore, the urban-rural divide should be taken into account in diabetes education practice. Rural patients tend to be younger, have less formal education, smoke and drink more, have shorter diagnosed disease duration, but are as likely as urban patients to develop diabetes complications as shown in our study. Given the exiting evidence on the association between socio-demographic factors and participation in diabetes education [46-47], various educational approaches tailored to the local socio-economic context will be essential to strengthen the capacity of diabetes care. Diabetes education with printed materials may help maximise the effective communication of health messages [48-49], which may be relevant to T2DM patients at higher CV risk who tend to have lower educational attainment in the urban-rural fringe.
Please see lines 324-328, 329-331, and 334-337 for the revision. Thank you again for all your valuable comments.